# AN EFFICIENT UNSUPERVISED FRAMEWORK FOR CONVEX QUADRATIC PROGRAMS VIA DEEP UNROLLING

## ABSTRACT

Quadratic programs (QPs) arise in various domains such as machine learning, finance, and control. Recently, learning-enhanced primal-dual hybrid gradient (PDHG) methods have shown great potential in addressing large-scale linear programs; however, this approach has not been extended to QPs. In this work, we focus on unrolling "PDQP", a PDHG algorithm specialized for convex QPs. Specifically, we propose a neural network model called "PDQP-net" to learn optimal QP solutions. Theoretically, we demonstrate that a PDQP-net of polynomial size can align with the PDQP algorithm, returning optimal primal-dual solution pairs. We propose an unsupervised method that incorporates KKT conditions into the loss function. Unlike the standard learning-to-optimize framework that requires optimization solutions generated by solvers, our unsupervised method adjusts the network weights directly from the evaluation of the primal-dual gap. This method has two benefits over supervised learning: first, it helps generate better primal-dual gap since the primal-dual gap is in the objective function; second, it does not require solvers. We show that PDQP-net trained in this unsupervised manner can effectively approximate optimal QP solutions. Extensive numerical experiments confirm our findings, indicating that using PDQP-net predictions to warm-start PDQP can achieve up to 45% acceleration on QP instances. Moreover, it achieves 14% to 31% acceleration on out-of-distribution instances.

## 1 INTRODUCTION

Convex *quadratic programs* (QPs) involve identifying a solution within the feasible region defined by linear constraints, aiming to minimize a convex quadratic objective function. This type of problem arises in various domains, including machine learning (Cortes, 1995; Candes et al., 2008), finance (Markowitz, 1952; Boyd et al., 2017), and control engineering (Garcia et al., 1989).

Extensive efforts have been devoted to developing efficient algorithms for convex QPs, with the most classic approaches being the *simplex* (Dantzig, 2016) and *barrier* (Andersen et al., 2003) algorithms. While both methods have shown robust performance in solving QPs, their scalability is often hindered by the computational overhead associated with matrix factorization, particularly in large-scale scenarios (Wright, 1997). To address this challenge, there has been growing interest in leveraging matrix-free *first-order methods* (FOMs) for optimizing convex QPs, such as SCS (O'Donoghue, 2021), OSQP (Stellato et al., 2020), and PDQP (Lu & Yang, 2023). Both OSQP and SCS are operator-splitting approaches that still require at least one iteration of matrix factorization, whereas PDQP is a truly matrix-free method that solves QPs by alternately utilizing primal and dual gradients. Consequently, PDQP shows greater potential for solving large-scale QPs. However, even with PDQP, tackling these problems typically necessitates thousands of iterations.

Recently, *machine learning* has been extensively utilized to expedite optimization algorithms (Bengio et al., 2021; Gasse et al., 2022; Chen et al., 2024a). Many works have trained *graph neural networks* (GNNs) to predict key problem properties, while theoretical research by Chen et al. (2024b) has shown that GNNs can reliably predict essential features of convex QP problems, such as feasibility, optimal objective value, and optimal solution. In practice, Li et al. (2024) introduces an unrolling approach that integrates GNNs with a *primal-dual hybrid gradient* (PDHG) solver called "PDLP", achieving efficient solution mappings for large-scale LPs. To our knowledge, no existing works have extended this line of research to address convex QPs. Furthermore, the commonly used end-to-end

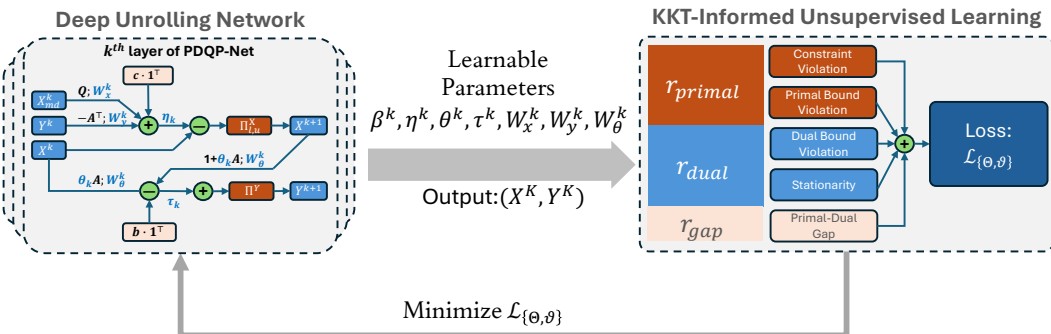

Figure 1: Overview of the proposed unsupervised learning framework. The left panel illustrates the architecture of the PDQP-Net, which is based on the algorithm-unrolling approach, while the right panel presents the KKT-informed unsupervised learning scheme.

approaches focus on minimizing the loss between predicted solutions and target labels, falling under the *supervised learning* paradigm. In addition to the high costs associated with label collection, the supervised framework faces two additional drawbacks: (i) multiple optimal solutions can introduce non-negligible noise during the training stage; (ii) the predicted primal-dual solution may deviate slightly from the optimal one while resulting in a significant duality gap. The latter issue is particularly pronounced for QPs due to the quadratic terms in the objective function.

In this work, we introduce a deep unrolling-based learning framework for efficiently solving convex QPs. We unroll the iterative PDQP algorithm into a GNN architecture termed "PDQP-Net", integrating optimality conditions—such as the Karush-Kuhn-Tucker (KKT) conditions—into the loss function for training. A complete demonstration of the proposed unsupervised learning framework is shown in Figure 1. During inference, PDQP-Net can predict near-optimal solutions for input QP instances, which serve as an effective initial solution for warm-starting the PDQP solver. The key contributions of our work can be summarized as follows.

- **PDQP-Net Architecture.** We introduce an algorithm-unrolled PDQP-Net that accurately replicates the PDQP algorithm. Our theoretical analysis demonstrates a precise alignment between PDQP-Net and the PDQP algorithm. Moreover, we prove that a PDQP-Net with $\mathcal{O}\left(\log \frac{1}{\epsilon}\right)$ neurons can approximate the optimal solution of QP problems to within an $\epsilon$-accuracy.

- **Unsupervised Framework.** We underscore the limitations of using supervised frameworks for learning QP solutions and propose an unsupervised training approach, leveraging a KKT-informed loss function.

- **Empirical Performance.** Our extensive experiments show substantial improvements over both supervised learning and traditional GNN-based methods. Specifically, warm-starting PDQP achieves up to a $45\%$ speedup and a $31\%$ performance boost on instances from various distributions.

## 2 PRELIMINARIES

### 2.1 QUADRATIC PROGRAMS

In this work, we consider a convex QP in the following form:

$$\min_{l \leq x \leq u} \quad \frac{1}{2}x^\top Q x + c^\top x$$
$$\text{s.t.} \quad Ax \geq b, \tag{1}$$

where $x \in \mathbb{R}^n$ represents the decision variable, $Q \in \mathbb{S}_+^n$ is a positive semi-definite matrix, and $c \in \mathbb{R}^n$ represents the linear coefficients. The constraints are specified by $A \in \mathbb{R}^{m \times n}$ and $b \in \mathbb{R}^m$.

## 2.2 THE PDQP ALGORITHM

One approach to solving problem (1) is to reformulate it as a *saddle point problem*:

$$\min_{l \le x \le u} \max_{y \ge 0} \mathcal{L}(x, y) = \frac{1}{2} x^\top Q x + c^\top x - y^\top (Ax - b), \quad (2)$$

where $y \ge 0$ are the dual variables corresponding to the inequality constraints. To optimize problem (2), first-order methods (FOMs) have gained traction due to their scalability and relatively low per-iteration computational cost, making them particularly effective for large-scale problems. Among these methods, the recently introduced restarted accelerated PDHG algorithm, termed PDQP (Li et al., 2024), exploits several acceleration techniques to achieve linear convergence. The PDQP algorithm is outlined in Algorithm 1. In essence, PDQP alternates between updates to the primal variable $x$ and the dual variable $y$, leveraging gradient information. The projection operators $\text{Proj}_{[l,u]}^x$ and $\text{Proj}_{[0,+\infty]}^y$, used in Step 6 and Step 7, ensure that the variables remain within their respective bounds. This approach avoids the need for solving linear systems, thereby bypassing the computational bottleneck of matrix factorization—a key advantage when tackling large-scale QPs.

---

**Algorithm 1** PDQP

---

1: **Input:** $Q, A, b, c, l, u$
2: **Initialize:** $x^0, y^0$ to all zero vectors
3: **Let:** $\bar{x}^0 = x^0$
4: **for** $k \in \{0, \dots, K\}$ **do**
5: $\quad x_{md}^k = \left(1 - \beta^k\right) \bar{x}^k + \beta^k x^k$
6: $\quad x^{k+1} = \text{Proj}_{[l,u]}^x \{x^k - \eta_k \left(Q x_{md}^k + c - A^\top y^k\right)\}$ $\qquad \qquad \triangleright$ primal update
7: $\quad y^{k+1} = \text{Proj}_{[0,+\infty]}^y \left\{y^k + \tau_k \left(b - A \left(\theta^k \left(x^{k+1} - x^k\right) + x^{k+1}\right)\right)\right\}$ $\qquad \triangleright$ dual update
8: $\quad \bar{x}^{k+1} = \left(1 - \beta^k\right) \bar{x}^k + \beta^k x^{k+1}$
9: $\quad$ **if** residuals $\le \epsilon$ **then** $\qquad \qquad \qquad \qquad \qquad \triangleright$ check optimality conditions
10: $\quad \quad$ Break
11: $\quad$ **end if**
12: **end for**
13: **Output:** $x^K, y^K$

---

## 3 KKT-INFORMED PDQP UNROLLING

While PDQP offers strong computational efficiency, the inherent long-tail behavior of FOMs often requires a large number of iterations for convergence. To mitigate this issue, we propose enhancing its efficiency by incorporating machine learning techniques. Specifically, our goal is to develop a deep learning framework capable of predicting the solutions generated by PDQP. Motivated by this, we introduce an unsupervised deep unrolling framework to solve convex QPs more efficiently. This section outlines the framework in three stages: the algorithm-unrolled PDQP-Net (Section 3.1), challenges of learning optimal QP solutions through supervised learning (Section 3.2), and the unsupervised learning framework we propose to overcome these challenges (Section 3.3).

### 3.1 UNROLLING THE PDQP ALGORITHM

In this part, we discuss the design of PDQP-Net by unrolling the PDQP algorithm to ensure alignment between the two. Previous unrolling approaches have primarily focused on specific real-world applications, such as compressive sensing (Zhang & Ghanem, 2018; Xiang et al., 2021) and image processing (Liu et al., 2019; Li et al., 2020). These methods demonstrate considerable performance improvements over conventional "black-box" neural networks and provide enhanced interpretability. To follow this trajectory, two critical factors must be considered when designing the neural network that unrolls the PDQP algorithm: learnable parameters and projection operators.

**Learnable parameters**. Building on the unrolling framework proposed for PDLP (Li et al., 2024), we incorporate a channel expansion technique. Specifically, at iteration $k$, the primal and dual variables, $x^k \in \mathbb{R}^n$ and $y^k \in \mathbb{R}^m$, are expanded into matrices $X^k \in \mathbb{R}^{n \times d_x^k}$ and $Y^k \in \mathbb{R}^{m \times d_y^k}$. This expansion is achieved by expressing $x^k$ and $y^k$ as linear combinations of $X^k$ and $Y^k$, using learnable parameters

$W_x^k$ and $W_y^k$, respectively. This design not only allows the network to generalize across varying problem sizes but also enhances its expressive capacity through the increased width of the network. Additionally, since step sizes (e.g., $\beta^k, \eta^k, \tau^k, \theta^k$ in Algorithm 1) are essential for the convergence of PDQP, they are dynamically adjusted at each iteration and replaced with scalar learnable parameters to better suit the network structure.

**Projection operators**. As seen in Step 6 and Step 7 of Algorithm 1, PDQP utilizes two projection operators on $x^k$ and $y^k$ to ensure that these variables respect their prescribed bounds throughout the optimization process. However, directly implementing such operators in neural networks can overlook unbounded variables and may lead to inefficient GPU computation.

Since these projections exhibit piece-wise linear behavior, we can derive new projection operators using shift and ReLU functions to define $\Pi_{[l,u]}^x$ and $\Pi_{[0,\infty]}^y$, which are more compatible with the neural network architecture:

$$\Pi_{[l,u]}^x(x) = x + \mathcal{I}_l \cdot \text{ReLU}\left(l - (x - \mathcal{I}_u \cdot \text{ReLU}(x - u))\right),$$
$$\Pi_{[0,+\infty]}^y(y) = \text{ReLU}(y),$$

where $\mathcal{I}_l$ and $\mathcal{I}_u$ are binary vectors indicating whether $x$ has lower or upper bounds, respectively.

By incorporating these elements, we introduce the PDQP-unrolled network architecture, PDQP-Net, as described in Algorithm 2. The input to the network consists of the QP instance $\mathcal{M} := (Q, A, b, c, l, u)$. Starting with zero-initialized vectors $x^0$ and $y^0$, the $k$-layer PDQP-Net—comprising learnable parameters $\Theta := \left(\beta^k, \eta^k, \theta^k, \tau^k, W_{\bar{x}}^k, W_y^k, W_\theta^k\right)$ and multi-layer perceptrons (MLPs) $f_x$, $f_y$, $g_x$, and $g_y$—produces predictions for the optimal solutions $x^*$ and $y^*$.

---

**Algorithm 2** PDQP-Net

---

1: **Input:** $\mathcal{M} := (Q, A, b, c, l, u)$
2: **Learnable parameters:** $\Theta := (\beta^k, \eta^k, \theta^k, \tau^k, W_{\bar{x}}^k, W_y^k, W_\theta^k)$, MLPs: $f_x, f_y, g_x, g_y$
3: **Initialize:** $x^0, y^0$ all zero vectors
4: **Let:** $X^0 \leftarrow f_x(x^0), Y^0 \leftarrow f_y(y^0), \bar{X}^0 \leftarrow X^0$                    ▷ initial embedding
5: **for** $k \in \{0, \ldots, K\}$ **do**
6:     $X_{md}^k = \left(1 - \beta^k\right)\bar{X}^k + \beta^k X^k$
7:     $X^{k+1} = \Pi_{[l,u]}^X\left\{X^k - \eta_k\left(QX_{md}^k W_{\bar{x}}^k + c \cdot \mathbf{1}_x^\top - A^\top Y^k W_y^k\right)\right\}$       ▷ primal update
8:     $Y^{k+1} = \Pi_{[0,\infty]}^Y\left\{Y^k + \tau_k\left(b \cdot \mathbf{1}_y^\top - A\left(\theta^k\left(X^{k+1} - X^k\right) + X^{k+1}\right)W_\theta^k\right)\right\}$    ▷ dual update
9:     $\bar{X}^{k+1}\left(1 - \beta^k\right)\bar{X}^k + \beta^k X^{k+1}$
10: **end for**
11: **Output:** 1-dimension vectors $x^* \leftarrow g_x(X^K), y^* \leftarrow g_y(Y^K)$

---

The following theorem guarantees that our proposed PDQP-Net can effectively align with the PDQP algorithm.

**Theorem 3.1.** *Given any QP instance $\mathcal{M} := (Q, A, b, c, l, u)$ and its corresponding primal-dual sequence $(x^k, y^k)_{k \leq K}$ generated by the PDQP algorithm within $K$ iterations, there exists a $K$-layer PDQP-Net with parameter assignment $\Theta_{PDQP}$ that can output the same iterative solution sequence.*

Theorem 3.1 establishes that with an appropriate configuration of parameters, PDQP-Net can replicate the sequence of solutions generated by the PDQP algorithm. This capability is crucial for ensuring that the learned network maintains both efficiency and consistency in capturing the algorithm's behavior across diverse problem instances. In addition to reproducing the solution sequence, we present a proposition that proves the convergence properties of PDQP-Net. By aligning its architecture with the PDQP algorithm, PDQP-Net inherits the same convergence characteristics, guaranteeing its efficacy in solving QPs.

**Proposition 3.1.** *Given an approximation error bound $\epsilon$, there exists a $K$-layer PDQP-Net with $\mathcal{O}\left(\log \frac{1}{\epsilon}\right)$ neurons that exhibits at least the same convergence properties as the PDQP algorithm. Furthermore, the predicted solutions of this network achieve linear convergence to an optimal solution.*

Proposition 3.1 confirms that PDQP-Net is not only capable of replicating the PDQP algorithm but also retains the algorithm's linear convergence rate. The design of learnable parameters, including

step sizes and projection operators, ensures that PDQP-Net remains efficient and reliable, converging to optimal solutions within a finite number of iterations.

**Remark 3.1.** *We note that PDQP-Net falls within the broader class of message-passing GNNs. In recent years, GNNs have gained considerable attention for their success in learning the solutions of optimization problems (Nair et al., 2020; Li et al., 2024). Despite the empirical success, Chen et al. (2022) and Chen et al. (2024b) take an initial step towards a theoretical understanding of the power of GNNs. They demonstrated that given sufficient parameters, GNNs can approximate the solution mappings of LPs and QPs to arbitrary precision. Their theoretical framework is based on the universal approximation theorem for MLP and Lusin's theorem, which guarantees the approximability of measurable functions. However, this framework does not establish a direct relationship between the performance of GNNs and the size of the network parameters. Achieving a desired level of approximation accuracy may still need an impractically large number of parameters.*

*In contrast, PDQP-Net is built on a more refined theoretical foundation. By algorithm unrolling, we prove that PDQP-Net can approximate the optimal solution of QP to an $\epsilon$-accuracy using only $\mathcal{O}\left(\log \frac{1}{\epsilon}\right)$ neurons. This result provides insight into how efficient and reliable convergence can be achieved with a given network size, thereby advancing our understanding of GNNs' representational power in solving QP problems.*

### 3.2 SUPERVISED LEARNING AND PRIMAL-DUAL GAPS

In the realm of end-to-end learning for solving optimization problems, models are typically trained using a supervised learning approach, as emphasized in recent studies (Li et al., 2024; Chen et al., 2024b). In such settings, the associated loss function $\mathcal{L}$ is often designed to quantify the discrepancy between the predicted solution and the optimal one. However, in addition to the high costs associated with label collection, the supervised framework faces another significant issue: the predicted primal-dual solution may deviate slightly from the optimal one while resulting in a substantial duality gap.

To illustrate this issue, we present the following proposition.

**Proposition 3.2.** *Let $(x^*, y^*)$ denote the optimal primal-dual solution to a given convex QP, and let $(x_0, y_0)$ represent any primal-dual solution. The normalized primal-dual gap $r_{gap}(x_0, y_0)$ is bounded as follows:*

$$r_{gap}(x_0, y_0) \leq \|Q\| \cdot \|x_0 - x^*\|^2 + \|c\| \cdot \|x_0 - x^*\| + \|b\| \cdot \|y_0 - y^*\| + R,$$

*where $R$ represents the difference in the reduced-cost contribution between $(x_0, y_0)$ and the optimal solution $(x^*, y^*)$.*

Proposition 3.2 indicates that the upper bound on the duality gap at $(x_0, y_0)$ depends not only on the solution discrepancy but also on problem characteristics such as $\|Q\|$, $\|c\|$, and $\|b\|$. Depending on the characteristics of $Q$, $c$, and $b$, even when the predicted primal-dual solution $(x_0, y_0)$ is close to the optimal one, the duality gap can still be significant. This issue is particularly pronounced for QPs, as the discrepancy can be amplified by the quadratic matrix $Q$.

### 3.3 KKT-INFORMED UNSUPERVISED LEARNING

To address the limitations identified in Section 3.2, it is crucial to integrate the primal-dual gap into the design of learning-based optimization algorithms. However, directly incorporating the primal-dual gap term into the existing supervised learning framework presents challenges, primarily because it does not align naturally with the numerical scale of the distance to the optimal solution. As a result, effectively embedding the primal-dual gap into the loss function necessitates a comprehensive redesign. Given the overarching objective of predicting solutions that demonstrate both optimality and feasibility, we propose substituting the distance to the optimal solution with a combination of metrics that simultaneously assess both aspects. One promising approach is to adhere to the optimality conditions outlined in Lu & Yang (2024), which incorporate a KKT-based metric that evaluates solution quality by accounting for both feasibility violations and the primal-dual gap. This method not only aligns with our objectives but also offers theoretical guarantees based on KKT conditions. Drawing inspiration from this, we derive a KKT-informed loss function that integrates considerations of both feasibility and optimality. Specifically, for a QP problem (1), we introduce its KKT conditions as follows:

- **Primal Feasibility:** $Ax - b \geq 0, l \leq x \leq u$
- **Dual Feasibility:** $y, \lambda, \mu \geq 0$
- **Stationary Condition:** $Qx + c - A^\top y - \lambda + \mu = 0$
- **Complementarity slackness:** $y^\top(Ax - b) + \lambda^\top(x - l) + \mu^\top(u - x) = 0$

where $\lambda$ and $\mu$ denote the dual variables associated with lower and upper bounds for $x$, respectively.

Since the QPs under consideration are convex, the KKT conditions are both necessary and sufficient for optimality. Minimizing the violation quantities associated with $x$ and $y$ will lead to the optimal solution. Building on the approach proposed by Lu & Yang (2024), we employ the primal-dual gap (as shown in Equation 3) as a measure, rather than applying complementarity slackness directly. This choice is well-justified, as the primal-dual gap becomes equivalent to the complementarity slackness when the stationary condition is met.

$$r_{gap} := x^\top Q x + c^\top y - y^\top b - \lambda^\top l + \mu^\top u \tag{3}$$

Based on this, we propose an unsupervised learning loss comprising three components: (i) *primal residual* ($r_{primal}$), which quantifies the violation of constraints and primal variable bounds; (ii) *dual residual* ($r_{dual}$), which assesses the violation of stationary conditions and dual variable bounds; and (iii) *primal-dual gap* ($r_{gap}$), which evaluates the optimality gap. Note that all three quantities are normalized to relative values, ensuring they are on the same numerical scale, which allows for direct summation. For the ease of distinguishing, we denote normalized quantities with $\hat{\cdot}$. The input to this loss function comprises the predicted primal solution $x^*$, the dual solution $y^*$, and the QP instance $\mathcal{M}$. Denoting the learnable parameters of involved MLPs as $\vartheta$, the loss function seeks to minimize the total residual with respect to the parameters $\{\Theta, \vartheta\}$. Specifically, for each QP instance $\mathcal{M}$, the unsupervised learning loss can be expressed as:

$$\mathcal{L}_{\{\Theta, \vartheta\}}(\mathcal{M}, x^*, y^*) := \widehat{r}_{primal} + \widehat{r}_{dual} + \widehat{r}_{gap}.$$

A detailed derivation procedure of this loss function can be found in Appendix H, along with the actual implementation in Appendix G.

According to Proposition 3.2, the supervised learning approach can potentially yield solutions with large primal-dual gaps, as it is rarely possible for a model to produce exact predictions in practice. As a result, employing this loss function is expected to yield higher-quality solutions compared to the supervised method. Additionally, utilizing an unsupervised learning approach eliminates the need for collecting optimal solutions for training—an extremely time-consuming task for complex QPs—thus enhancing sample complexity.

## 4 EXPERIMENTS

In this section, we present comprehensive numerical results to evaluate the performance of our proposed unsupervised framework. We begin by showcasing PDQP-Net's ability to predict high-quality solutions and compare these results with those from a supervised framework, highlighting the advantages of an unsupervised approach in end-to-end settings. Next, we demonstrate how these solutions can empirically accelerate the original PDQP algorithm. We also assess the performance improvements achieved through the algorithm-unrolling PDQP-Net by comparing it with a GNNs-based method. Additionally, we explore the framework's capacity to learn a unified algorithm and generalize to out-of-distribution instances. Moreover, we validate our hypothesis that even when the distance to the optimal solution is small, the primal-dual gap can still be significant, leading to increased PDQP iterations. Finally, a time profiling is also included to demonstrate the scalability of our proposed framework.

### 4.1 EXPERIMENT SETTINGS

**Dataset.** Our numerical experiments are primarily conducted on three datasets: QPLIB (Furini et al., 2018), synthetic random QPs, and for the out-of-distribution task, the Maros–Mészáros dataset (Maros & Mészáros, 1999). Detailed data splitting and instance specifications are provided in the Appendix D.

**Metric.** In order to evaluate the effectiveness of different approaches, we utilize the optimality conditions introduced in Section 3.3 to measure the quality of the generated solution. Additionally, we employ the acceleration improvement (Improv.) to assess the acceleration by warm-starting PDQP with solutions generated by the proposed framework (ours), which is computed as Improv. $:= \frac{\text{PDQP}-\text{ours}}{\text{PDQP}}$.

**Training Protocol.** For implementation, we used PyTorch 2.0.1 (Paszke et al., 2019) for developing the neural networks and PDQP.jl (Lu & Yang, 2024) to implement for warm-starting the PDQP algorithm. The models were trained using the AdamW optimizer, with an initial learning rate of $1e^{-4}$ and a maximum of 1,000 iterations. All experiments were conducted on a server equipped with two NVIDIA V100 GPUs, an Intel(R) Xeon(R) Gold 5117 CPU @ 2.00GHz, and 256GB of RAM.

## 4.2 SUPERVISED-LEARNING VS UNSUPERVISED-LEARNING

In this section, we demonstrate the efficiency of the proposed PDQP-Net framework in producing high-quality predictions and compare its performance to that of a supervised learning approach. Specifically, we present primal residual ($\hat{r}_{primal}$), dual residual ($\hat{r}_{dual}$) and primal-dual gap ($\hat{r}_{gap}$) of the predicted solutions on both synthetic instances and real-world instances from QPLIB. Smaller residuals indicate better performance.

The consolidated results are presented in Table 1. It is evident from the table that PDQP-Net consistently produces solutions with small relative residuals. Notably, the relative residuals and gaps of solutions generated by the unsupervised approach are consistently below 10% across all datasets. Given the inherent limitations of first-order methods (FOMs) in precision, these results underscore the relatively high quality of PDQP-Net's predictions.

Table 1: Results of comparing the proposed framework against the supervised learning one. We report the relative residuals of each approach and the problem sizes of each dataset. Better results are highlighted with **bold font**.

| dataset. | size | Unsupervised | | | Supervised | | |
|---|---|---|---|---|---|---|---|
| | (n/m) | $\hat{r}_{primal} \downarrow$ | $\hat{r}_{dual} \downarrow$ | $\hat{r}_{gap} \downarrow$ | $\hat{r}_{primal} \downarrow$ | $\hat{r}_{dual} \downarrow$ | $\hat{r}_{gap} \downarrow$ |
| QPLIB-8845 | (1,546/777) | **0.0417** | **0.0002** | **0.0414** | 0.6429 | 0.4650 | 0.5794 |
| QPLIB-3547 | (1,998/3,137) | **0.0490** | **0.0041** | **0.0066** | 0.4308 | 0.0041 | 0.0068 |
| QPLIB-8559 | (10,000/5,000) | **0.0884** | **0.0** | **0.0015** | 0.3653 | 0.0 | 1.2052 |
| SYN-small | (1,000/1,000) | 0.0418 | **0.0** | **0.0223** | **0.0227** | 0.0 | 0.1559 |
| SYN-mid | (5,000/5,000) | 0.0392 | **0.0** | **0.0155** | **0.0219** | 0.0002 | 0.1166 |
| SYN-large | (5,000/20,000) | **0.0034** | **0.0** | **0.0069** | 0.0073 | 0.0 | 0.055 |

Furthermore, a comparative analysis of solution quality between our framework and the supervised approach demonstrates the advantages of the proposed unsupervised approach in this task. As shown in the table, the supervised method struggles to close the primal-dual gap, reinforcing our argument that merely learning the optimal solution does not ensure a model capable of accurately predicting high-quality primal and dual solution pairs. In contrast, the unsupervised framework not only bypasses the need for labor-intensive data collection, but also consistently produces more closely aligned primal and dual solutions. Specifically, on the QPLIB-8559 dataset, the unsupervised approach predicts solutions with a primal-dual gap of 0.0015, which is approximately 1,000 times smaller than those produced by the supervised model. Additionally, on challenging instances from QPLIB, our framework consistently delivers solutions with superior feasibility. This improvement can largely be attributed to our specifically designed loss function, which simultaneously optimizes for both feasibility and optimality, allowing PDQP-Net to outperform the supervised approach in terms of solution quality and convergence.

These findings highlight the effectiveness of the proposed unsupervised approach, especially in challenging problem settings, and further validate its ability to produce high-quality, feasible solutions.

## 4.3 THE EFFECT OF WARM-STARTING PDQP

In this section, we evaluate the empirical benefits of using the proposed framework by examining the acceleration it brings to PDQP by warm-starting PDQP by PDQP-Net's predictions. In Table 2, we present the solving time and the number of iterations for three different approaches: **1)** the default PDQP (PDQP), **2)** warm-started PDQP with the supervised approach (PDQP+SL), and **3)** warm-started PDQP with the unsupervised approach (PDQP+UL). Additionally, we report the improvement ratio (Improv.) of PDQP+UL over the default PDQP solver.

The reported improvement ratios demonstrate a substantial acceleration of PDQP+UL compared to the default PDQP, primarily due to the high-quality solutions for warm-starting PDQP. Specifically, the overall acceleration exceeds $30\%$, with the SYN-small dataset achieving an improvement ratio of $45\%$. A similar trend is observed in the reduction of the number of iterations, where PDQP+UL reduces iterations by up to $49.05\%$. Moreover, PDQP+UL consistently outperforms both the original PDQP and PDQP+SL in terms of time and iterations across all datasets. These results, when considered alongside the aforementioned experiments, confirm that warm-starting PDQP with solutions with smaller primal-dual gaps leads to more significant acceleration.

These results confirm that PDQP+UL, by producing high-quality solutions with smaller primal-dual gaps, could significantly improve solving time and reduce iterations across all datasets. A time profiling of this experiment is also included in Appendix E to validate the scalability of our proposed framework.

Table 2: Results of warm-starting PDQP by using the predicted solutions. We report the time and number of iterations for solving instances via different approaches. Improvement ratio is also reported. Better results are highlighted with **bold font**.

| Dataset | Time (sec.) | | | | # Iterations | | | |
|---|---|---|---|---|---|---|---|---|
| | PDQP | PDQP+SL | PDQP+UL | Improv.↑ | PDQP | PDQP+SL | PDQP+UL | Improv.↑ |
| QPLIB-8845 | 104.51 | 89.61 | **76.17** | 30.64% | 182338.9 | 161547.6 | **135566.7** | 25.38% |
| QPLIB-3547 | 3.90 | 2.41 | **2.34** | 40.06% | 1840.7 | 1728.9 | **1726.5** | 6.20% |
| QPLIB-8559 | 181.14 | 141.52 | **118.89** | 34.37% | 143947.8 | 136745.0 | **130760.9** | 9.16% |
| SYN-small | 7.89 | 6.12 | **4.30** | 45.54% | 1008.0 | 664.0 | **513.6** | 49.05% |
| SYN-mid | 13.16 | 11.91 | **9.01** | 31.64% | 1067.67 | 838.0 | **697.0** | 34.69% |
| SYN-large | 319.02 | 227.55 | **217.87** | 31.70% | 12167.2 | 8754.5 | **8674.6** | 28.71% |

## 4.4 COMPARING PDQP-NET AGAINST GNNS

To evaluate the performance of our algorithm-unrolling network, we conducted comparative experiments against conventional GNNs (Chen et al., 2024b). The quality of solutions was assessed using three key metrics: primal residuals ($\hat{r}_{primal}$), dual residuals ($\hat{r}_{dual}$), and the relative gap ($\hat{r}_{gap}$), with lower values indicating better performance. The best results are highlighted in bold.

As shown in Table 3, the unrolled PDQP-Net consistently outperforms conventional GNNs across nearly all residuals and datasets. Notably, PDQP-Net demonstrates significant advantages on the QPLIB benchmarks, consistently producing lower primal and dual residuals, along with smaller relative gaps. For instance, on the QPLIB-8845 dataset, PDQP-Net achieves a primal residual of $0.0417$ and a dual residual of $0.0002$, compared to the $0.6473$ and $0.4706$ reported by the GNN model. This superior performance extends to the QPLIB-3547 and QPLIB-8559 datasets, further underscoring the robustness of PDQP-Net. Beyond the QPLIB benchmarks, we evaluated both models on synthetic datasets of various sizes. PDQP-Net continued to deliver lower residuals, particularly excelling on the SYN-large dataset, showcasing its scalability in tackling larger and more complex problems. Moreover, the performance gain is evident not only in the unsupervised PDQP-Net but also in its supervised variant, validating the architectural improvements over traditional GNNs. These results emphasize PDQP-Net's effectiveness in generating high-quality solutions, reinforcing its superiority over conventional GNNs.

Table 3: Results of comparing the proposed framework against a GNNs-based approach. We report the relative residuals of each approach and the problem sizes of each dataset. Better results are highlighted with **bold font**.

| Dataset | size | PDQP-Net | | | GNNs | | |
|---|---|---|---|---|---|---|---|
| | (n/m) | $\hat{r}_{primal} \downarrow$ | $\hat{r}_{dual} \downarrow$ | $\hat{r}_{gap} \downarrow$ | $\hat{r}_{primal} \downarrow$ | $\hat{r}_{dual} \downarrow$ | $\hat{r}_{gap} \downarrow$ |
| QPLIB-8845 | (1,546/777) | **0.0417** | **0.0002** | **0.04114** | 0.6473 | 0.4706 | 0.5852 |
| QPLIB-3547 | (1,998/3,137) | **0.0490** | **0.0041** | **0.0066** | 0.5035 | 0.0041 | 0.1290 |
| QPLIB-8559 | (10,000/5,000) | **0.0884** | **0.0** | **0.0015** | 0.4015 | 0.0 | 1.4598 |
| SYN-small | (1,000/1,000) | 0.0418 | **0.0** | **0.0223** | **0.0334** | 0.0 | 0.1494 |
| SYN-mid | (5,000/5,000) | 0.0392 | **0.0** | **0.0155** | **0.225** | 0.0 | 0.1042 |
| SYN-large | (5,000/20,000) | **0.0034** | **0.0** | **0.0069** | 0.0054 | 0.0 | 0.0717 |

## 4.5 GENERALIZING TO QPS FROM DIVERSE DISTRIBUTIONS

Our approach, which unrolls an existing general algorithm and minimizes its residuals, has the potential to learn a generic function capable of handling various problems. In this section, we evaluate this potential by training and testing the proposed framework on the Maros–Mészáros dataset (Maros & Mészáros, 1999), which comprises problems from diverse applications.

Table 4 presents the solving times for PDQP and warm-started PDQP, along with the acceleration ratios achieved by warm-starting PDQP. We report these numbers on 5 randomly selected test intances. The results demonstrate that the model, trained on out-of-distribution instances, is able to generate high-quality predictions that significantly accelerate PDQP. Approximately, a 20% average acceleration is observed, with a significant 31.61% speedup on the QSHIP04L problem, making our approach nearly 20 times faster than the default PDQP. However, on the DUAL4 dataset, warm-starting PDQP did not result in any significant acceleration, likely due to the small size of the problem.

These findings suggest that our framework has the potential to be applied in more generalized scenarios where problems come from different distributions. Nevertheless, the failure case indicates that the model's ability to generalize to out-of-distribution instances may still be limited, possibly due to an insufficient number of parameters.

Table 4: Results on acceleration via warm-starting PDQP, where the test set is from a different distribution from the training set.

| Instance | QSHIP04L | QISREAL | CVXQP2_M | QBRANDY | DUAL4 |
|---|---|---|---|---|---|
| PDQP time (sec.) | 17.84 | 2.34 | 1.36 | 6.29 | 2.04 |
| PDQP+WS time (sec.) | 12.20 | 1.72 | 1.17 | 5.40 | 2.09 |
| Improv. | 31.61% | 26.31% | 16.38% | 14.21% | $-2.12\%$ |

## 4.6 UNDERSTANDING WHY THE UNSUPERVISED APPROACH WORKS

**Distance vs primal-dual gap** In Proposition 3.2, we show that a solution can exhibit a large primal-dual gap even when it is closer to the optimal one. Figure 2 illustrates this phenomenon by plotting the primal-dual gap of multiple randomly perturbed points against their distance to the optimal solution for the datasets QPLIB-8559, QPLIB-3547 and QPLIB-8845. Each dot represents a perturbed solution.

The results reveal that, for both problems, even when the solution has nearly zero distance from the optimum, the primal-dual gap can still be significant. Furthermore, as the distance from the optimum increases, the primal-dual gap rises significantly, often at an exponential rate. These observations reinforce the notion that a large primal-dual gap can occur even for near-optimal solutions, which explains why predictions given by supervised-learning has large residuals.

**Unsupervised learning achieves better gap** In this section, we demonstrate that our proposed unsupervised learning approach significantly reduces the primal-dual gap. Figure 3 illustrates the

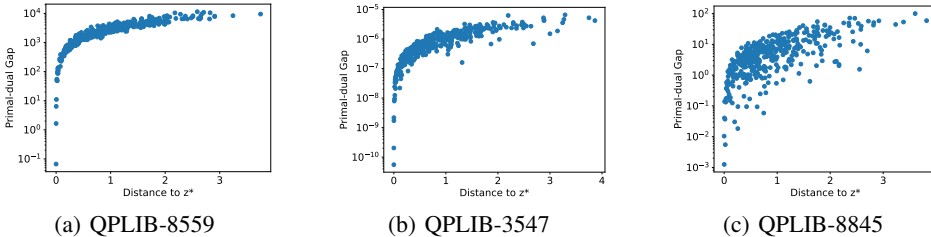

(a) QPLIB-8559     (b) QPLIB-3547     (c) QPLIB-8845

Figure 2: Plot of the primal-dual gap of points generated by randomly perturbing the optimal solution, using the $\ell_2$ distance to measure the deviations. Each dot represents a perturbed solution. The figure illustrates the relationship between the perturbation distance and the corresponding primal-dual gap on two QPLIB instances, demonstrating solutions with small distances could have large primal-dual gap.

relationship between primal-dual gaps (measured by $l_2$ norm) and distances to the optimal solution for each iteration during the training process of both supervised and unsupervised frameworks. Each dot represents an iteration.

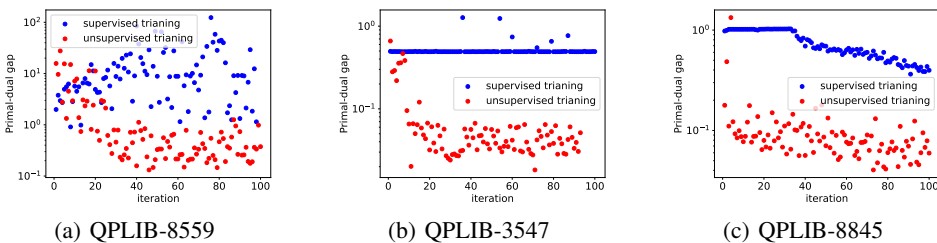

(a) QPLIB-8559     (b) QPLIB-3547     (c) QPLIB-8845

Figure 3: Plot of the primal-dual gap of each iteration during the training process of the supervised and the unsupervised framework, using the $\ell_2$ norm to compute the gap. Each dot represents an iteration. The figure demonstrates that the unsupervised framework always achieves better primal-dual gap over the supervised one.

The plot reveals that the iterations from the unsupervised approach consistently achieve much better primal-dual gaps compared to those from supervised learning. Additionally, the points from unsupervised learning cluster at a very low gap level, whereas supervised learning struggles to efficiently close the gap. These observations support the conclusion that the unsupervised learning approach is more effective at generating primal-dual solution pairs with smaller gaps, ultimately providing high-quality predictions that can efficiently accelerate PDQP.

## 5 CONCLUSIONS

This work presents a unsupervised learning framework for obtaining optimal solutions to QPs. Central to this framework is a KKT-condition-based loss function that effectively integrates optimality conditions to improve solution predictions. We also introduce the PDQP-Net architecture, which unrolls the original PDQP algorithm, providing a compelling alternative to traditional GNN backbones. Our theoretical analysis shows that the unsupervised framework effectively addresses the challenges of supervised learning, particularly the issue of large primal-dual gaps even when the distance to the optimal solution is minimal. Furthermore, our proposed PDQP-Net can replicate the original PDQP algorithm, inheriting its convergence properties while also enhancing expressiveness through channel expansion. Comprehensive numerical experiments further validate our approach, showing substantial improvements in prediction quality and empirical acceleration of the PDQP algorithm. These findings highlight the potential of integrating optimization theory into the L2O paradigm, suggesting that such integration can substantially enhance the performance of existing optimization algorithms.

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

## A PROOF OF THEOREM 3.1

We prove the existence of a $K$-layer PDQP-Net that, with a specific parameter assignment $\Theta_{PDQP}$, replicates the sequence of primal-dual updates generated by the PDQP algorithm for any QP instance $\mathcal{M} := (Q, A, b, c, l, u)$.

The PDQP algorithm generates iterates $(x^k, y^k)_{k \leq K}$ by updating $x^k$ and $y^k$ iteratively, starting from initial points $x^0$ and $y^0$. The updates at each iteration $k$ are as follows:

$$x^{k+1} = \text{Proj}^x_{[l,u]} \left( x^k - \eta_k \left( Qx^k_{md} + c - A^\top y^k \right) \right),$$

$$y^{k+1} = \text{Proj}^y \left( y^k + \tau_k \left( b - A \left( \theta(x^{k+1} - x^k) + x^{k+1} \right) \right) \right),$$

where

$$x^k_{md} = (1 - \beta^k)\bar{x}^k + \beta^k x^k,$$

and $\bar{x}^k$ is the momentum term:

$$\bar{x}^{k+1} = (1 - \beta^k)\bar{x}^k + \beta^k x^{k+1}.$$

The PDQP-Net is a neural network designed to mimic the above iterative updates through learnable parameters $\Theta_{PDQP} = \{\beta^k, \eta^k, \tau^k, W^k_{\bar{x}}, W^k_x, W^k_y\}$ for $k \leq K$. The PDQP-Net generates embedding sequences $(X^k, Y^k)_{k \leq K}$ where:

$$X^{k+1} = \Pi^X_{[l,u]} \left( X^k - \eta_k \left( QX^k_{md}W^k_{\bar{x}} + c\mathbf{1}^\top_x - A^\top Y^k W^k_y \right) \right),$$

$$Y^{k+1} = \Pi^Y \left( Y^k + \tau_k \left( b\mathbf{1}^\top_y - A \left( \theta(X^{k+1} - X^k) + X^{k+1} \right) W^k_x \right) \right),$$

and

$$X^k_{md} = (1 - \beta^k)\bar{X}^k + \beta^k X^k,$$

with $\bar{X}^k$ defined analogously to $\bar{x}^k$ in PDQP:

$$\bar{X}^{k+1} = (1 - \beta^k)\bar{X}^k + \beta^k X^{k+1}.$$

We proceed by induction to show that at each iteration $k$, there exists a set of parameter assignments $\Theta_{PDQP}$ for the PDQP-Net such that:

$$X^k = x^k \quad \text{and} \quad Y^k = y^k.$$

**Base Case.** At iteration $k = 0$, both the PDQP and PDQP-Net initialize $x^0 = 0$ and $y^0 = 0$, resulting in $X^0 = 0$ and $Y^0 = 0$. Thus, the base case holds trivially.

**Inductive Step.** Assume that for some $k \geq 0$, $X^k = x^k$ and $Y^k = y^k$. We now show that $X^{k+1} = x^{k+1}$ and $Y^{k+1} = y^{k+1}$ for a suitable choice of parameters in $\Theta_{PDQP}$.

From the PDQP update, we know:

$$x^{k+1} = \text{Proj}^x_{[l,u]} \left( x^k - \eta_k(Qx^k_{md} + c - A^\top y^k) \right),$$

and

$$y^{k+1} = \text{Proj}^y \left( y^k + \tau_k \left( b - A(\theta(x^{k+1} - x^k) + x^{k+1}) \right) \right).$$

The corresponding PDQP-Net updates are:

$$X^{k+1} = \Pi^X_{[l,u]} \left( X^k - \eta_k \left( QX^k_{md}W^k_{\bar{x}} + c\mathbf{1}^\top_x - A^\top Y^k W^k_y \right) \right),$$

$$Y^{k+1} = \Pi^Y \left( Y^k + \tau_k \left( b\mathbf{1}^\top_y - A \left( \theta(X^{k+1} - X^k) + X^{k+1} \right) W^k_x \right) \right).$$

By setting $W^k_{\bar{x}} = W^k_x = W^k_y = I$ (the identity matrix) and ensuring that $\eta_k$, $\tau_k$, and $\beta_k$ in PDQP-Net match the corresponding step sizes in PDQP, it is straightforward to verify that:

$$X^{k+1} = x^{k+1} \quad \text{and} \quad Y^{k+1} = y^{k+1}.$$

Thus, by induction, the PDQP-Net can exactly replicate the primal-dual sequence generated by the PDQP algorithm for all iterations $k \leq K$.

This completes the proof that for any QP instance $\mathcal{M} := (Q, A, b, c, l, u)$, there exists a parameter assignment $\Theta_{PDQP}$ such that a $K$-layer PDQP-Net can output the same primal-dual sequence $(x^k, y^k)_{k \leq K}$ as the PDQP algorithm.

## B    PROOF OF PROPOSITION 3.1

We aim to demonstrate that there exists a $K$-layer PDQP-Net with a specific parameter assignment that exhibits at least the same convergence properties as the PDQP algorithm, including linear convergence of its predicted solutions to an optimal solution of the given QP problem.

The PDQP algorithm is well-known for its linear convergence under standard conditions such as strong convexity of the objective function and appropriate assumptions on the constraints. Specifically, the primal-dual iterates $(x^k, y^k)$ converge to an optimal solution $(x^*, y^*)$ at a linear rate. To extend this convergence guarantee to PDQP-Net, we will demonstrate that, with a carefully selected parameter assignment, the predicted solutions $(X^k, Y^k)$ from PDQP-Net will converge similarly.

The updates in the PDQP-Net for the primal-dual iterates $(X^k, Y^k)$ are given by:

$$X^{k+1} = \Pi^X_{[l,u]} \left( X^k - \eta_k \left( QX^k_{md}W^k_{\bar{x}} + c\mathbf{1}^\top_x - A^\top Y^k W^k_y \right) \right),$$

$$Y^{k+1} = \Pi^Y \left( Y^k + \tau_k \left( b\mathbf{1}^\top_y - A \left( \theta(X^{k+1} - X^k) + X^{k+1} \right) W^k_\theta \right) \right),$$

where the momentum term is:

$$X^k_{md} = (1 - \beta^k)\bar{X}^k + \beta^k X^k.$$

To further ensure that PDQP-Net achieves linear convergence, we now leverage results from PDQP. In particular, we rely on the convergence guarantees provided by Theorem 3.1, which states that a sequence generated by the PDQP algorithm exhibits linear convergence under appropriate conditions. This theorem forms the foundation of our analysis.

**Theorem B.1.** *[Lu & Yang (2024)] (Linear Convergence): Consider the sequence $\{z_n\}^\infty_{n=0}$ generated by the PDQP algorithm with fixed restart frequency for solving the following quadratic programming problem (QP):*

$$\min_x \quad \frac{1}{2}x^\top Qx + c^\top x \quad \text{subject to} \quad Ax = b,$$

*where $Q \in \mathbb{R}^{n \times n}$ is a positive semidefinite matrix, and $A \in \mathbb{R}^{m \times n}$ defines the equality constraints. Suppose that, for any $\xi > 0$, the objective function satisfies quadratic growth with parameter $\alpha_\xi$ on a ball $B_R(z_0)$ centered at $z_0$ with radius*

$$R = \frac{3}{1 - 1/e}dist(z_0, Z^*),$$

*where $Z^*$ is the set of optimal solutions. Then, for each inner iteration $0 \leq k \leq K - 1$, the parameters $\beta_k$, $\theta_k$, $\eta_k$, and $\tau_k$ are chosen as follows:*

$$\beta_k = \frac{k+2}{2}, \quad \theta_k = \frac{k}{k+1}, \quad \eta_k = \frac{k+1}{2(\|Q\| + K\|A\|)}, \quad \tau_k = \frac{k+1}{2K\|A\|}.$$

*Furthermore, if the restart frequency $K$ satisfies*

$$K \geq \max\left(\frac{32e^2\|Q\|}{\alpha_\xi}, \frac{32e^2\|A\|}{\alpha_\xi}, \frac{64\|Q\|}{\xi}, \frac{64\|A\|}{\xi}, \frac{\|Q\|}{\|A\|}\right),$$

*then for any outer iteration $n$, the following holds:*

$$(i) \quad \|z_n - z_0\| \leq \frac{3}{1 - 1/e} dist(z_0, Z^*),$$

$$(ii) \quad dist(z_n, Z^*) \leq e^{-n} dist(z_0, Z^*).$$

This result shows that we achieve an $\epsilon$-close solution to the QP in the sense of distance to optimality after:

$$O\left(\max\left(\frac{\|Q\|}{\alpha_\xi}, \frac{\|A\|}{\alpha_\xi}, \frac{\|Q\|}{\xi}, \frac{\|A\|}{\xi}, \frac{\|Q\|}{\|A\|}\right) \log \frac{dist(z_0, Z^*)}{\epsilon}\right)$$

iterations.

Using the insights from Theorem B.1, we now turn to PDQP-Net. To ensure that PDQP-Net exhibits the same linear convergence as PDQP and PDQP, we assign the parameters $\eta_k$, $\tau_k$, and $\beta_k$ in PDQP-Net to mirror those in the PDQP algorithm. Specifically, we choose:

$$\eta_k = \frac{k+1}{2(\|Q\| + K\|A\|)}, \quad \tau_k = \frac{k+1}{2K\|A\|},$$

and set the weight matrices $W_{\bar{x}}^k$, $W_x^k$, and $W_y^k$ as identity matrices, ensuring that the updates in PDQP-Net closely follow the PDQP algorithm's structure. This parameter choice guarantees that the primal-dual iterates $(X^k, Y^k)$ of PDQP-Net exhibit linear convergence.

With this parameter assignment, we can conclude that the predicted solutions $(X^k, Y^k)$ of PDQP-Net converge linearly to the optimal solution $(X^*, Y^*)$ of the QP problem. Specifically, for sufficiently small $\epsilon > 0$, the distance to optimality is bounded by:

$$\|X^k - X^*\| + \|Y^k - Y^*\| \leq C\rho^{k(\epsilon)},$$

where $C > 0$ is a constant and $\rho \in (0, 1)$ is the linear convergence rate. This bound directly follows from the conditions outlined in Theorem B.1,.

We only need to take $k(\epsilon) = \lceil \frac{1}{C} \log_\rho \epsilon \rceil + 1$, then we get

$$\|X^k - X^*\| + \|Y^k - Y^*\| \leq \epsilon.$$

Since the network width is a fixed number and $\rho \in (0, 1)$, the number of neurons at each layer is bounded by a constant. Thus, the total number of neurons has the same order as $K(\epsilon)$, which is $\mathcal{O}\left(\log \frac{1}{\epsilon}\right)$. By assigning the parameters of PDQP-Net according to the framework of Theorem 1, we ensure that the predicted solutions of the $K$-layer PDQP-Net converge linearly to the optimal solution, matching the convergence properties of both the PDQP and PDQP algorithms.

## C   PROOF OF PROPOSITION 3.2

*Proof.* As discussed earlier, the primal-dual gap is a crucial metric for assessing solution quality. Assuming stationary conditions in $r_{dual}$ is satisfied, we can express the equivalent primal objective function as $P(x) = c^\top x + 0.5x^\top Qx$ and the dual objective function as $D(y) = b^\top y - 0.5x^\top Qx$.

Consequently, the complete formulation of $r_{gap}$ can be expressed as follows:

$$r_{gap}(x_0, y_0) = |P(x_0) - D(y_0) - \mathbf{RCC}(x_0, y_0)| = |c^\top x_0 - b^\top y_0 + x_0^\top Q x_0 - \mathbf{RCC}(x_0, y_0)|$$

The procedure of deriving $\mathbf{RCC}$ parallels that of the $\mathbf{RCV}$ term in $r_{dual}$, which is detailed in Appendix H. Since $(x^*, y^*)$ are the optimal solutions, we can assume that the primal-dual gap $r_{gap}(x^*, y^*) = 0$.

$$r_{gap}(x_0, y_0) = |P(x_0) - D(y_0) - \mathbf{RCC}(x_0, y_0)| - |P(x^*) - D(y^*) - \mathbf{RCC}(x^*, y^*)|$$

We can then derive the following:

$$= |P(x_0) - D(y_0) - \mathbf{RCC}(x_0, y_0) - (P(x^*) - D(y^*) - \mathbf{RCC}(x^*, y^*))|$$

by decomposing $r_{gap}(x_0, y_0)$:

$$r_{gap}(x_0, y_0) = |(P(x_0) - P(x^*)) - (D(y_0) - D(y^*)) - (\mathbf{RCC}(x_0, y_0) - \mathbf{RCC}(x^*, y^*))|$$

Using the triangle inequality, we can write:

$$r_{gap}(x_0, y_0) \le |P(x_0) - P(x^*)| + |D(y_0) - D(y^*)| + |\mathbf{RCC}(x_0, y_0) - \mathbf{RCC}(x^*, y^*)|$$

We now bound each term separately. For the primal term, using a second-order Taylor expansion:

$$|P(x_0) - P(x^*)| \le \|c\| \cdot \|x_0 - x^*\| + \frac{1}{2}\|Q\| \cdot \|x_0 - x^*\|^2$$

For the dual term:

$$|D(y_0) - D(y^*)| \le \|b\| \cdot \|y_0 - y^*\| + \frac{1}{2}\|Q\| \cdot \|x_0 - x^*\|^2$$

The RCC contains information about the upper and lower bounds of the decision variables, which vary from instance to instance, and we note $|\mathbf{RCC}(x^*, y^*) - \mathbf{RCC}(x_0, y_0)|$ as $R$. And For the reduced cost correction term, we denote the deviation as:

$$|\mathbf{RCC}(x_0, y_0) - \mathbf{RCC}(x^*, y^*)| = R$$

Thus, combining these bounds:

$$r_{gap}(x_0, y_0) \le \|c\| \cdot \|x_0 - x^*\| + \|Q\| \cdot \|x_0 - x^*\|^2 + \|b\| \cdot \|y_0 - y^*\| + R$$

Finally, simplifying the expression and introducing the constants $C'_x = \|Q\|$, $C_x = \|c\|$, and $C_y = \|b\|$, we arrive at the desired bound:

$$r_{gap}(x_0, y_0) \le C'_x \|x_0 - x^*\|^2 + C_x \|x_0 - x^*\| + C_y \|y_0 - y^*\| + R$$

This concludes the proof.

$\square$

## D  DATASET

In this section, we present the detailed settings of each utilized datasets.

**QPLIB (Furini et al., 2018)** This dataset includes a diverse collection of QP instances. We selected several convex QPs with linear constraints and relaxed any integer variables. To facilitate training for each problem, we generate new instances by randomly perturbing the coefficients. These instances are then split into training and testing sets at a 9-to-1 ratio. Detailed sizes and $\gamma$ are reported in Table 5, along with data splitting information.

**Maros & Mészáros (1999)** This smaller QP dataset comprises 138 instances from various domains. In Table 6, we present sizes of tested instances.

Table 5: Detailed information of utilized QPLIB instances

| Instance | # vars. | # cons. | # nnz. | $\gamma$ | # train | # test |
|---|---|---|---|---|---|---|
| QPLIB-8845 | 1,546 | 777 | 10,999 | 0.1 | 450 | 50 |
| QPLIB-3547 | 1,998 | 3,137 | 8,568 | 0.1 | 450 | 50 |
| QPLIB-8559 | 10,000 | 5,000 | 24,998 | 0.1 | 450 | 50 |

Table 6: Detailed information of utilized Maros–Mészáros instances

| Instance | # vars. | # cons. | # nnz. |
|---|---|---|---|
| QSHIP04L | 2,118 | 402 | 6,33 |
| QISREAL | 142 | 174 | 2269 |
| CVXQP2_M | 1,000 | 250 | 749 |
| QBRANDY | 249 | 220 | 2,148 |
| DUAL4 | 75 | 1 | 75 |

**Random QP (SYN)** In order to validate the performance of the proposed framework on large-scale instances, we generated random QPs in the following form:

$$\min_{x} \quad \frac{1}{2}x^\top D x + c^\top x$$
$$\text{s.t.} \quad Ax \geq b$$
$$0 \leq x \leq u$$

To ensure convexity, we set $D$ as a diagonal matrix with positive entries, while $c$ is randomly sampled from a normal distribution. The matrix $A$ is generated to have a specified density $\rho$, with its entries also drawn from a particular normal distribution. To enhance the feasibility of the generated instances, we define $b_i = \alpha(a_i^\top u)$, where $\alpha$ is a control parameter for feasibility. Table 7 provides detailed sizes and the distributions used for generating this synthetic dataset

Table 7: Detailed information of generating synthetic instances

| Instance | # vars. | # cons. | $\rho$ | $\alpha$ | D | c | A |
|---|---|---|---|---|---|---|---|
| SYN-small | 1,000 | 1,000 | 0.3 | 0.8 | $\sim \mathcal{N}(4,2)$ | $\sim \mathcal{N}(3,1)$ | $\sim \mathcal{N}(2,1)$ |
| SYN-mid | 5,000 | 5,000 | 0.1 | 0.8 | $\sim \mathcal{N}(4,2)$ | $\sim \mathcal{N}(3,1)$ | $\sim \mathcal{N}(2,1)$ |
| SYN-large | 5,000 | 20,000 | 0.05 | 0.99 | $\sim \mathcal{N}(4,2)$ | $\sim \mathcal{N}(3,1)$ | $\sim \mathcal{N}(2,1)$ |

## E    TIME PROFILING

In this section, we demonstrate the scalability of our proposed framework by reporting the inference time (Inf. time) required to generate predictions, and comparing it to the solving time (Sol. time) of PDQP after warm-starting the PDQP solver. In Table 8, we also provide the ratio between inference and solving times (Inf./Sol. ratio) for a clearer comparison. Additionally, the number of non-zeros (NNZ) is reported to indicate problem sizes.

The results indicate that the inference time is negligible relative to the solving time, with the inference-to-solving ratio remaining below $1\%$ across all datasets. Moreover, the inference time does not increase significantly with the number of non-zeros, demonstrating the framework's efficiency in scaling to larger problems. These trends highlight the framework's ability to effectively handle datasets of larger scales with minimal computational overhead.

## F    VALIDATION OF PROJECTION OPERATORS

In Section F˜ H, we provide details on the implementation. Empirically, dual variables can be unbounded when they correspond to equality constraints. To handle these cases, we introduce another

Table 8: Inference time versus solving time after warm-starting PDQP. NNZ and inference/solving time ratio are also included.

|  | QPLIB-8845 | QPLIB-3547 | QPLIB-8559 | SYN-small | SYN-mid | SYN-large |
|---|---|---|---|---|---|---|
| NNZ | 10,247 | 8,556 | 14,998 | 299,905 | 2,500,773 | 5,002,401 |
| Inf. time (sec.) | 0.0138 | 0.0131 | 0.0645 | 0.0160 | 0.0876 | 0.2144 |
| Sol. time (sec.) | 76.17 | 2.34 | 118.89 | 4.30 | 9.01 | 217.87 |
| Inf./Sol. ratio | $< 1e^4$ | 0.5% | $< 1e^4$ | 0.37% | 0.97% | $< 1e^4$ |

binary vector extracted from the QP, denoted as $\mathcal{I}_y$. Specifically, $\mathcal{I}_y$ is a binary vector that indicates whether a given element of $y$ corresponds to an inequality constraint.

In this part, we validate that the projection operator can recover those utilized in the original PDQP algorithm.

$$\Pi^x_{[l,\infty]}\{x\} = \quad x + \mathcal{I}_l. * \text{ReLU}(l - x)$$

**Case1** $\quad x \geq l \Rightarrow l - x \leq 0 : \Pi^x_{[l,\infty]}\{x\} = x$

**Case2** $\quad x \leq l \Rightarrow l - x \geq 0 : \Pi^x_{[l,\infty]}\{x\} = x + l - x = l$

**Case3** $\quad x$ unbounded $: \Pi^x_{[l,\infty]}\{x\} = x$

$$\Pi^x_{[-\infty,u]}\{x\} = \quad x - \mathcal{I}_u. * \text{ReLU}(x - u)$$

**Case1** $\quad x \geq u \Rightarrow x - u \geq 0 : \Pi^x_{[-\infty,u]}\{x\} = x - x + u = u$

**Case2** $\quad x \leq u \Rightarrow x - u \leq 0 : \Pi^x_{[-\infty,u]}\{x\} = x$

**Case3** $\quad x$ unbounded $: \Pi^x_{[-\infty,u]}\{x\} = x$

$$\Pi^x_{[l,u]}\{x\} = \quad x + \mathcal{I}_l. * \text{ReLU}\left(l - (x - \mathcal{I}_u. * \text{ReLU}(x - u))\right)$$

**Case1** $\quad x \geq u \Rightarrow x - u \geq 0 : \Pi^x_{[l,u]}\{x\} = x - x + u = u$

**Case2** $\quad l \leq x \leq u \Rightarrow x - u \leq 0 : \Pi^x_{[l,u]}\{x\} = x$

**Case2** $\quad x \leq l \Rightarrow l - x \geq 0 : \Pi^x_{[l,u]}\{x\} = x + l - x = l$

**Case4** $\quad x$ unbounded $: \Pi^x_{[l,u]}\{x\} = x$

$$\Pi^y_{[0,\infty]}\{y\} = \quad y + \mathcal{I}_y. * \text{ReLU}(-y)$$

**Case1** $\quad y \geq 0 : \Pi^y_{[0,\infty]}\{y\} = y$

**Case2** $\quad x \leq 0 : \Pi^y_{[0,\infty]}\{y\} = y - y = 0$

## G  IMPLEMENTATION OF LOSS FUNCTION

For the implementation of the unsupervised loss, variable bounds of primal variable $l \leq x \leq u$ and dual variable $y \geq 0$ should be considered. With the bounds indicator binary vectors $\mathcal{I}_y$, $\mathcal{I}_l$, and $\mathcal{I}_u$, we focus on three different types of residuals:

**Primal Residual** ($\hat{r}_{primal}$): This component of the loss function primarily aims to maximize the feasibility of primal variables by addressing both bounds and constraint violations. The value is normalized by a non-gradient-propagating term, $\|Ax\|_\infty$.

- Bounds violation: $\mathbf{BV} = \text{ReLU}(l - x) \cdot \mathcal{I}_l + \text{RelU}(x - u) \cdot \mathcal{I}_u$
- Constraints violation: $\mathbf{CV} = Ax - b + \text{ReLU}(b - Ax) \cdot I_y$
- $\hat{r}_{primal} = \frac{\|\mathbf{BV};\mathbf{CV}\|_\infty}{\max(\|b\|_\infty, \|Ax\|_\infty)}$

**Dual Residual** ($\hat{r}_{dual}$): This component ensures dual feasibility by addressing both the feasibility of dual variables and the stationary of the saddle point problem regarding $x$. The value is also normalized by non-gradient-propagating terms, $\|A^\top y\|_\infty$ and $\|Qx\|_\infty$.

- Primal gradient: $\zeta = c - A^\top y + Qx$

- stationary: $\mathbf{RCV} = \zeta$ - $\text{RelU}(\zeta)\,\mathcal{I}_l$ - $\min(0,\zeta)\,\mathcal{I}_u$
- Dual variable violation: $\mathbf{DV} = \text{RelU}(-y, 0.0)$ is_inequality
- $\hat{r}_{dual} = \frac{\|\mathbf{RCV};\mathbf{DV}\|_\infty}{\epsilon + max(\|c\|_\infty, \|Qx\|_\infty, \|A^\top y\|_\infty)}$

**Primal-Dual Gap ($\hat{r}_{gap}$)** As discussed earlier, the primal-dual gap is a crucial metric for assessing solution quality. Assuming stationary in $\hat{r}_{dual}$ is satisfied, we can express the equivalent primal objective function as $P = c^\top x + 0.5x^\top Q x$ and the dual objective function as $D = b^\top y - 0.5x^\top Qx$. However, since stationary may not always hold during the optimization process, we introduce a new term, $\mathbf{RCC}$ to address this issue. Consequently, the complete formulation of $\hat{r}_{gap}$ can be expressed as follows:

- $\mathbf{RC} = RelU(\zeta)\mathcal{I}_l + min(0,\zeta)\mathcal{I}_u$
- $\mathbf{RCC} = \sum_{i:\mathbf{RC}_i>0} l_i\mathbf{RC}_i + \sum_{i:\mathbf{RC}_i<0} u_i\mathbf{RC}_i$
- $\hat{r}_{gap} = \frac{|P-D-\mathbf{RCC}|}{\epsilon+\max(P,D)} = \frac{|c^\top x - b^\top y + x^\top Qx - \mathbf{RCC}|}{\epsilon+\max(P,D)}$

Here, the primal-dual gap is equivalent to complementarity slackness if and only if stationary is satisfied. However, in the primal-dual gap $P - D$ formulation, we do not account for the objective contributions of the bounds. This contribution can be easily computed using the reduced cost, represented by the $\mathbf{RCC}$ term. The procedure of deriving $\mathbf{RCC}$ parallels that of the $\mathbf{RCV}$ term in $\hat{r}_{dual}$, which is detailed in Appendix H.

Finally, we conclude the final loss function by computing the summation of these three terms. This approach is valid because all values are normalized to the same scale.

# H  DERIVATION OF DUAL RESIDUAL

In this section, we explain how we derive $\mathbf{RCV}$ in the dual residual. We begin by presenting the saddle point problem, which incorporates the lower bounds $l$ and upper bounds $u$ for variables $x$:

$$\min_{l \leq x \leq u} \max_{y \geq 0, \lambda \geq 0, \mu \geq 0} \quad \frac{1}{2}x^\top Qx + c^\top x - (Ax - b)^\top y - (x - l)^\top \lambda - (u - x)^\top \mu.$$

Its gradient regarding $x$ is:
$$\nabla_x = Qx + c - A^\top y - \lambda + \mu.$$

For the sake of simplicity, we denote $Qx + c - A^\top y$ as $\zeta$. To ensure stationary, it is necessary for $\zeta = 0$. Consequently, each variable can have one of four scenarios:

1. $-\infty < x_i < \infty$:
$$\zeta_i = 0 \Rightarrow Vio = |\zeta|$$

2. $-\infty < x < u$:
$$\zeta_i = -\mu_i \leq 0 \Rightarrow \text{Violation:} |\max(\zeta_i, 0)| = |\zeta_i - \min(\zeta_i, 0)|$$

3. $l < x < \infty$:
$$\zeta_i = \lambda_i \geq 0 \Rightarrow \text{Violation:} |\min(\zeta_i, 0)| = |\zeta_i - \max(\zeta_i, 0)|$$

4. $l < x < u$:
$$\zeta_i = \lambda_i - \mu_i \Rightarrow \text{Violation:} |\zeta_i - \min(\zeta_i, 0) - \max(\zeta_i, 0)| = 0$$

Using the binary vectors $\mathcal{I}_u$ and $\mathcal{I}_l$ to represent variables with upper and lower bounds, respectively, we can express the above violation in a generalized form as follows:

$$\|\zeta - \max(\zeta, 0)\mathcal{I}_l - \min(0, \zeta)\mathcal{I}_u\|,$$

which is $\mathbf{RCV}$ in the proposed dual residual loss.

