# OpenReview forum: "An Efficient Unsupervised Framework for Convex Quadratic Programs via Deep Unrolling"
_ICLR.cc/2025/Conference — ICLR 2025 Conference Withdrawn Submission_

### Official Review · Reviewer_m3is · 2024-10-28

**Soundness:** 2
**Presentation:** 1
**Contribution:** 2
**Rating:** 3
**Confidence:** 3

**Summary:**

This paper proposes an efficient unsupervised framework to accelerate the solving of convex quadratic programs (QPs) using the unrolling technique. In particular, the authors propose architectures mimicking the PDQP algorithm (possibly standing for "Primal-Dual Quadratic Programming"; its meaning is never explicitly mentioned in the paper), termed as PDQP-net, and an unsupervised training loss based on the KKT optimality conditions. Through theoretical and empirical argument, they show that not only does this proposal improve the primal-dual gap, but it also gets rid of the need for a solver, which might be potentially costly to obtain for large-scale QPs.

**Strengths:**

- The paper is of sufficient interest: Using deep learning techniques and architectures to accelerate optimization is an interesting idea; this paper belongs to this line of research and the problem it addresses is relevant for many communities in ICLR.
- Integrating the unsupervised approach into the framework as in this paper is interesting since we no longer need a solver to generate training data (which also reduces the data preparation time, consequently).

**Weaknesses:**

The presentation of this paper has several problems. Let me mention a few examples:
- Several acronyms are not explained: PDQP, LPs (line 51).
- A more detailed introduction is recommended for Section 2.2. This entire work is based on the so-called PDQP algorithm and the algorithm, its main intuition, and results have not been introduced thoroughly.
- In Algorithm 1, the IF condition in line 9 is not explained: the variable "residual" is not defined yet.
- In line162, please specify the dimension of $W^k_x$ and $W^k_y$. In general, I think it is a good habit to mention the dimensions of variables when they are first introduced.
- I suggest that Section 3.2 should be incorporated into Section 3.3: its content is not sufficiently rich enough to stand alone and its goal is to motivate why the authors propose an unsupervised loss function.
- In Section 3.3, the definitions of three loss functions $r_{primal}, r_{dual}$ and $r_{gap}$ should be introduced in the main text, and not the Appendix. Same remark for their normalized version.
- There is a double citation of the paper "A practical and optimal first-order method for large-scale convex quadratic programming" in lines 587 - 591.

Certain formulations and arguments in the paper need adjustment, in my opinion. Here are several examples:
- Proposition 3.2 and Section 3.2 are confusing. I think the authors want to convince us that the distance between the optimal solution and the predicted one is not a good training loss. This is debatable because in many situations such as image denoising, the distance is more important. Also, Proposition 3.2 does not seem to support their argument since it provides an upper bound.
- Theorem B.1 in Appendix B analyses a different formulation from (1). I do not read this proof carefully but the author should re-check it.
- In Appendix B, the training instances are generated by taking an instance from QPLIB, perturbing with Gaussian noise. However, the authors test with the same instances that they used to generate training data (see Tables 1 and 5). Is that fair?

**Questions:**

- Why do the authors to have box constraints on $x$ (i.e. $l \leq x \leq u$)? Can we remove this condition since $Ax \geq b$ already contains these box constraints?
- Why do the authors unroll PDQP, and not other algorithms for convex quadratic programming? With this framework, I believe any first-order methods can be unrolled, in the same manner. Or maybe PDQP is the only first-order method for convex quadratic programming.

---

### Official Review · Reviewer_yUEU · 2024-10-30

**Soundness:** 2
**Presentation:** 2
**Contribution:** 3
**Rating:** 5
**Confidence:** 4

**Summary:**

The paper introduces PDQP-Net, a novel framework that leverages deep unrolling and unsupervised learning to efficiently solve convex QP. The neural network structure mimics the PDQP algorithm, with each network layer corresponding to an iteration step of PDQP. The loss function is based on KKT conditions to achieve a lower primal-dual gap and better primal and dual feasibilities. Compared to the supervised MSE of the solution, the KKT-based loss does not require labels (optimal solutions), thus not only avoiding time-consuming data collection but also resulting in better optimality gaps.

For numerical experiments, PDQP-Net is faster than traditional first-order methods, PDQP, and could be an effective starting point for better PDQP convergence. In addition, PDQP-Net achieves strong out-of-distribution generalization.

**Strengths:**

1. **Novelty:** In my opinion, the primary contribution is the KKT-based loss function. This unsupervised approach not only removes the need for costly optimal solution data but also improves solution quality.
2.  **Efficiency:**: Using model outputs as a warm-start of traditional PDGH significantly accelerates QP.

**Weaknesses:**

1. The PDQP architecture, including deep unrolling, trainable parameters, projection operators, and channel expansion, and its use as a warm start to accelerate convergence closely mirrors existing work, PDLP. While effective, these techniques do not constitute novel contributions in the context of this paper, as they are directly aligned with the PDLP paper.
2. While the paper provides some foundational hyperparameter settings, such as the learning rate, optimizer, and training iterations, it lacks some details, such as the GNN architecture, the depth $K$, and the design of multi-layer perceptrons (MLPs) $f_x$, $f_y$, $g_x$, and $g_y$.
3. Although the experiments provide comparisons regarding acceleration as a warm start against traditional pure QDHG, the paper does not include a direct runtime comparison between the neural network model and the traditional QDHG. Without this data, it is difficult to evaluate the efficiency of the neural network and assess the trade-offs between efficiency and optimality.

**Questions:**

1. I like the KKT loss. The concept shares similarities with the approach used in Physics-Informed Neural Networks (PINNs), which incorporate physical laws as constraints in the loss function to guide unsupervised learning. I suggest discussing connections with PINNs in related work.
2. It is useful to report the feasibility ratio. Compared to the feasibility residual, the feasibility ratio would provide a more direct indication of reliability and effectiveness, as feasible solutions may hold little to no practical value.
3. Is there potential for using the approach for nonconvex QP? Since nonconvex QPs are commonly encountered in practical applications but present additional challenges in terms of solution quality and convergence, the approach also seems promising for nonconvex cases.
4. The section captain "Comparing PDQP-Net Against GNNs" could indeed be confusing, as PDQP-net itself is also based on a GNN. Additionally, “GNNs” alone does not clarify the baselines used in the comparison.
5. Apart from "Expressive Power of Graph Neural Networks for (Mixed-Integer) Quadratic Programs," there are other recent methods for solving (not only) convex QPs. For instance, "DC3: A Learning Method for Optimization with Hard Constraints" provides an unsupervised approach based on soft constraints penalty and projection techniques. Have the authors considered comparing PDQP-Net with additional learning baselines like DC3 to further validate its performance?
6. The paper briefly mentions the network depth $K$ corresponding to the number of unrolled layers. Could the authors provide more insight into how different values of $K$ impact solution quality and computational efficiency?

---

### Official Review · Reviewer_k5BN · 2024-10-31

**Soundness:** 3
**Presentation:** 3
**Contribution:** 1
**Rating:** 3
**Confidence:** 4

**Summary:**

This paper introduces PDQP-net, a neural network architecture designed to learn the solution mappings for convex QPs. Inspired by first-order methods, PDQP-net is capable of emulating the PDQP algorithm. The authors also propose an unsupervised training approach based on the KKT conditions for training PDQP-net. Experiments on large-scale instances demonstrate that PDQP-net achieves faster convergence and smaller primal/dual errors.

**Strengths:**

- The proposed PDQP-net architecture is well-motivated and grounded in the principles of first-order optimization methods.
- The unsupervised training approach based on KKT conditions is efficient in training PDQP-net without requiring ground-truth optimal solutions.
- The experimental results on large-scale instances showcase the superior performance of PDQP-net in terms of convergence speed and primal/dual errors.

**Weaknesses:**

- The technical contributions, such as the construction proof for neural network complexity, are incremental compared to previous work [1].
- While the construction proof demonstrates that a PDQP-net exists that can recover the PDQP algorithm and achieve linear convergence, the convergence guarantee for PDQP-net with trainable parameters remains unclear. Providing convergence guarantees for the training process is crucial for the unrolling-based L2O scheme [2].
- Given that the construction proof relies on the PDQP algorithm, which already achieves the optimal convergence rate among first-order methods, the authors should discuss the reasons behind the performance gain of PDQP-net. Theoretically, is there (constant factor) complexity improvement of PDQP-net compared with the PDQP algorithm?
- The author provides a complexity of PDQP-net as $O(log(1/\epsilon))$ in Prop. 3.1, but the dependence on the problem dimensions ($n$ and $m$) is unclear, which is also critical for understanding the scalability of this framework.

**Questions:**

- How does PDQP-net generalize across varying problem sizes (line 162), and are the trainable parameters invariant to the problem dimension?

- In the experiments, the number of layers (K) used in PDQP-net is not specified. The authors should provide details on the choice of K and analyze how the performance changes across different values of K, as this is crucial for understanding the convergence behavior of the unrolling structure. It would also be helpful to present the practical parameter complexity of PDQP-net in experiments, especially for large-scale instances (e.g., n=10,000).

- How about using recurrent neural networks to parameterize the learnable parameters at each layer [2], making the trainable parameters invariant to the number of layers?

- In Prop. 3.2, the authors claim that supervised training could lead to a large primal-dual gap, as shown in the upper bound. How about directly minimizing the upper bound in the supervised training scheme?



[1] Li, B., Yang, L., Chen, Y., Wang, S., Chen, Q., Mao, H., ... & Sun, R. PDHG-Unrolled Learning-to-Optimize Method for Large-Scale Linear Programming. In Forty-first International Conference on Machine Learning.

[2] Liu, J., Chen, X., Wang, Z., Yin, W., & Cai, H. Towards constituting mathematical structures for learning to optimize. In International Conference on Machine Learning.


I will adjust my scores if the concerns are addressed.

---

### Official Review · Reviewer_ssAW · 2024-11-04

**Soundness:** 1
**Presentation:** 2
**Contribution:** 2
**Rating:** 3
**Confidence:** 4

**Summary:**

This paper proposes a deep unrolling method for solving QP problems. Particularly, the authors choose a PDQP method (PDHG applied on QP), parameterize some components in PDQP, unroll PDQP and truncate it into finite iterations. The parameterized unrolled algorithm is then treated as a neural network and trained using a KKT-informed loss function. Finally, experiments are conducted to evaluate the performance of the proposed PDQP-net and to illustrate the effectiveness of the KKT loss function.

**Strengths:**

The KKT-informed training approach is an interesting idea, especially for primal-dual methods. It is simple, principle-informed, and does not require the optimal solution for training.

**Weaknesses:**

While the paper presents some interesting concepts, it feels somewhat incomplete.

1. The theoretical results are limited. Theorem 3.1 is straightforward: by setting parameters in the unrolled method to mimic PDQP, the conclusion follows. Additionally, Proposition 3.1 directly follows from the PDHG convergence results [1].

2. The experimental setup could be stronger. (i) For baseline comparisons, the proposed method is primarily compared with PDHG. A more recognized open-source solver is OSQP [2]. Additionally, as a first-order method, PDHG typically have very slow convergence for ill-conditioned problems (although its single-step complexity is cheap). More robust solutions like interior-point solvers (e.g., Gurobi) would be a stronger baseline. (ii) The dataset is insufficient to fully support the conclusions. For instance, the "QPLIB-8845" instance used in the paper is a single instance rather than a dataset [3], meaning the GNN is only trained on this one instance (along with some perturbed variations). If we want to solve another QP, we have to train another GNN for that. Typically, training GNN is more complex than solving a single QP, so we would want the trained GNN to generalize across multiple QP instances.

3. References on unrolling are lacking. Surprisingly, even the first paper on unrolling [4] is not discussed. A more comprehensive review of unrolling techniques would be beneficial.

[1] Chambolle and Pock. "A First-Order Primal-Dual Algorithm for Convex Problems with Applications to Imaging." Journal of Mathematical Imaging and Vision 2011.

[2] Stellato et al. "OSQP: an operator splitting solver for quadratic programs." Mathematical Programming Computation 2020.

[3] https://qplib.zib.de/QPLIB_8845.html

[4] Gregor and LeCun. "Learning fast approximations of sparse coding." ICML 2010.

**Questions:**

see "Weaknesses"

---

### Official Review · Reviewer_vSzF · 2024-11-04

**Soundness:** 3
**Presentation:** 3
**Contribution:** 2
**Rating:** 3
**Confidence:** 3

**Summary:**

This paper explores the application of a machine learning-enhanced approach for solving quadratic programs (QPs); the method extends the primal-dual hybrid gradient (PDHG) method from linear programming to QPs. The authors introduce "PDQP-net," a neural network model that integrates the PDHG algorithm for convex QPs. A key innovation in this work is the unsupervised training approach, which incorporates KKT conditions into the loss function, allowing the network to be updated based on primal-dual gap evaluation rather than requiring solutions from traditional solvers.
Specifically, the paper extends the PDHG method for linear programming problems introduced by "Bingheng Li et al, PDHG-Unrolled Learning-to-Optimize Method for Large-Scale Linear Programming"

**Strengths:**

- the presented idea to incorporate KKT conditions in the learning to optimize setting is intuitive and well-motivated
- the method is clearly described without ambiguity
- the paper is mathematically rigorous, although I did not verify the validity of the proofs of the theorems in this paper
- the experimental results of the paper are substantiated with large-scale QP problems

**Weaknesses:**

The novelty of the proposed idea seems to be the main issue.
- as far as I understand the method "deep unrolling" refers to the unrolled operations of the operator splitting method for successive projection operations for primal and dual variables. As far as I can see, this is equivalent to the ADMM method. However, there is no mention of prior work on ADMM or other operator splitting methods.
- There are already existing works on differentiable or "unrolled" operator splitting methods as layers in deep neural networks for solving optimization problems in the learning to optimize setting
- the paper would benefit from a related work section highlighting similarities and differences between recent learning to optimize methods for parametric QP problems
- specifically, authors are encouraged to include recent advancements in differentiable optimization layers including QP-layers ADMM layers, and machine learning-based warm starting and operator splitting methods for QP and NLP problems
- This method looks very similar to that presented in the paper:
"Differentiable Linearized ADMM" https://proceedings.mlr.press/v97/xie19c.html
- Other related work that uses differentiable Douglas-Rachford (DR) layers to warm-start QP is closely related but not mentioned in the paper
"End-to-End Learning to Warm-Start for Real-Time Quadratic Optimization"
https://proceedings.mlr.press/v211/sambharya23a.html
- limitations of the proposed method are missing

**Questions:**

- KKT conditions are known to be sensitive to small perturbations in the problem parameters, often leading to ill-conditioned problems. How are these issues avoided in the proposed method?
- How does this method relate to the ADMM method and other operator-splitting algorithms such as Douglas-Rachford (DR)?

---

### Note · Authors · 2024-11-24

I have read and agree with the venue's withdrawal policy on behalf of myself and my co-authors.